# Mobilising Cross-Sectoral Collaboration in Creating Age-Friendly Cities: Case Studies from Akita and Manchester

**DOI:** 10.3390/ijerph22010073

**Published:** 2025-01-08

**Authors:** Patty Doran, Sophie Yarker, Tine Buffel, Hisami Satake, Fumito Watanabe, Minoru Kimoto, Ayuto Kodama, Yu Kume, Keiko Suzuki, Sachiko Makabe, Hidetaka Ota

**Affiliations:** 1Sociology Department, University of Manchester, Manchester M13 9PL, UK; tine.buffel@manchester.ac.uk; 2Health Geography, School of Science, University of Salford, Salford M5 4WT, UK; s.k.yarker@salford.ac.uk; 3ALL-A Co., Ltd., Akita 010-0976, Japan; hisami-s@all-a.jp (H.S.); f-watanabe@all-a.jp (F.W.); 4Department of Physical Therapy, Akita University Graduate School of Health Sciences, Akita 010-8502, Japan; minoru-kimoto@hs.akita-u.ac.jp; 5Department of Occupational Therapy, Akita University Graduate School of Health Sciences, Akita 010-8502, Japan; ay-kodama@med.akita-u.ac.jp (A.K.); kume.yuu@hs.akita-u.ac.jp (Y.K.); 6Advanced Research Center for Geriatric and Gerontology (ARGG), Akita University, Akita 010-8502, Japan; hidetaka-ota@med.akita-u.ac.jp; 7Department of Nursing, Akita University Graduate School of Health Sciences, Akita 010-8502, Japan; keiko@hs.akita-u.ac.jp (K.S.); smaka@hs.akita-u.ac.jp (S.M.)

**Keywords:** ageing, cross-sectoral collaboration, age-friendly, case study, leadership, older people, co-production

## Abstract

Developing Age-Friendly Cities and Communities (AFCCs) is an increasingly popular policy response to supporting ageing populations. AFCC programmes rely on cross-sectoral collaboration, involving partnerships among diverse stakeholders working across sectors to address shared goals. However, there remains a limited understanding of what mechanisms and strategies drive collaboration among diverse actors within age-friendly cities. To address this gap, this empirical paper draws on examples from a comparative case study across Akita (Japan) and Manchester (UK), two cities with distinct demographic profiles but both with a longstanding commitment to the age-friendly approach. Case studies were created through a range of data collection methods, namely, a review of secondary data sources, semi-structured interviews with key stakeholders, and fieldwork in each city. Key insights from the case studies relating to the mobilisation of cross-sectoral collaboration were categorised into three themes: leadership and influencing, co-production, and place-based working. These mechanisms are not mutually exclusive; collaboration building through co-production and place-based working is essential to deliver age-friendly programmes, but these mechanisms rely on leadership and influence. Therefore, it is recommended that all three mechanisms be used to effectively mobilise cross-sectoral collaborations to collectively create AFCC and support healthy ageing.

## 1. Introduction

This paper provides insights into the question of how to develop Age-Friendly Cities and Communities (AFCCs), with a particular focus on the mechanisms and strategies driving cross-sectoral collaboration in contrasting urban contexts. AFCC are defined as environments that optimise opportunities for health, participation, and security for people as they age by providing accessible services, inclusive built environments, and opportunities for social connections [1]. The AFCC concept was developed in 2007 by the World Health Organization (WHO) through a study exploring how cities could become more inclusive for older people [1]. Building on this work, the WHO launched the Global Network for AFCC in 2010, which rapidly expanded and included over 1600 cities and communities across 53 countries by late 2024, with network members sharing a commitment to promoting active and healthy ageing through their age-friendly programmes. This expansion highlights the enduring significance of the age-friendly approach as a policy response to urban population ageing, further emphasised by the United Nation’s Decade of Healthy Ageing (2021–2030).

Alongside the growth of the Global Network, attention to AFCC has also increased within academic research [2]. Across multiple disciplines, studies have focused on the planning [3], implementation [4], monitoring, and evaluation [5,6] of age-friendly initiatives. Researchers have also examined the development of AFCC in diverse settings, encompassing both rural [7] and urban environments [8], as well as in different countries in the Global North and South [9,10]. The use of the ‘age-friendly’ construct has also expanded beyond cities and communities to include businesses, universities, and healthcare systems. Fulmer et al. [11] argue for the creation of an ‘age-friendly ecosystem’, where initiatives to improve the lived environment, the healthcare system and a prevention-focused public health system can create synergies and collectively contribute to fostering healthy ageing. Menec and Brown [12] (p. 2) conceptualise the AFCC movement as a community development approach: ‘the emphasis is on how local government and/or community members to work toward a community or city becoming more age-friendly’. This perspective aligns with Buffel and Phillipson’s [8] view of age-friendly work as a partnership-based approach involving various stakeholders in supporting ageing in place, including local authorities, community organisations, older people, and the private sector.

Despite the expansion of AFCC programmes and research, critical knowledge gaps persist regarding the dynamics of collaborative processes essential for achieving age-friendly change [13]. Specifically, questions remain as to ‘how to mobilise cooperation among diverse actors’ [14] (p. 59), how to establish ‘multisectoral action that delivers outcomes in ways that reduce inequities’ [6] (p. 2), and how to ensure that ‘older people [are] centrally involved’ in such partnerships [15] (p. 6). While Black & Oh [16], in their review of 30 AFCC programmes across the US, found strong evidence of multisectoral working as a defining feature of AFCC programmes, they also highlighted the need for further ‘in-depth research such as case studies … to better understand the *facilitative factors* among high-performing AFC collaborations’ [16] (emphasis added, p. 14).

Addressing this gap, this study explores the dynamics and strategies for mobilising cross-sectoral collaborations within the AFCC movement across diverse urban contexts, providing much-needed empirical insights into these collaborative processes. To conceptualise cross-sectoral collaboration, we draw on collaborative governance literature, which identifies trust, commitment and a shared understanding across all stakeholders as essential elements of effective collaboration [17]. Similarly, a study that explored the characteristics of collaborations between ageing professionals and planners found that trust, reputation, and reciprocity were essential for engaging different stakeholders in collaborative work [18]. Coalition building in urban environments is similarly driven by shared goals and alliances that align stakeholder objectives to create more inclusive and equitable cities [19]. Further, Fulmer and colleagues suggest uniting different sectors under a common agenda can help solve complex social challenges, such as those posed by the ageing population [20].

Building on these characteristics, this paper conceptualises cross-sectoral collaboration as the joint effort of diverse stakeholders—including local governments, community organisations, businesses, universities and older adults—to pool resources, share and leverage expertise, and create synergies to address the diverse needs of ageing populations. Such collaborations have been shown to be a key factor in the success of age-friendly programmes and initiatives [9,21]. As reflected in the AFCC framework, the challenges associated with ageing in cities cross between health, social, economic, and environmental domains. It has been argued that to meet these priorities and facilitate the necessary cross-sectoral partnerships requires leadership and/or a champion who can influence others to ‘come on board’ with the age-friendly agenda [11,22].

This study examines the mechanisms fostering cross-sectoral collaboration among diverse stakeholders in Akita (Japan) and Manchester (UK), two cities with distinct demographic, cultural and socioeconomic profiles yet sharing a longstanding commitment to collaborative age-friendly initiatives. Akita, traditionally an agricultural hub and now a city with nearly two in every five residents aged 60 or older, faces challenges associated with rapid population ageing and outward migration. In contrast, Manchester is a diverse and relatively young urban centre, where many older residents experience poverty and multi-morbidities. Despite these differences, both cities share a proactive approach to supporting their ageing populations, having joined the WHO Global Network of AFCC relatively early after its inception in 2010 and consistently pursuing age-friendly initiatives with financial support from their respective local governments. By selecting cities with contrasting demographic, cultural, and socio-economic contexts, this study enables an exploration of both the shared and context-specific collaborative processes that shape the implementation of AFCC initiatives in diverse settings.

Three years of collaborative work between Akita and Manchester researchers has enabled city comparisons, with regular meetings dedicated to co-designing all aspects of this study’s research framework, methodology, data collection and analysis. The resulting case studies unveil unique insights into the intricacies, mechanisms and challenges associated with fostering cross-sectoral collaboration for age-friendly change, revealing city-specific dynamics while highlighting essential transferable elements for effective implementation across diverse urban contexts. Two research questions were addressed: (1) Who are the key stakeholders engaged in fostering AFCC in Akita and Manchester? (2) What mechanisms drive cross-sectoral collaboration among diverse actors within these cities?

This paper presents, first, the history and development of age-friendly programmes in Akita and Manchester; second, the methodology of this study; and, third, findings with regard to the mechanisms of cross-sectoral collaboration in the two cities, these are grouped under three themes: leadership and influencing, co-production, and place-based working. The article concludes by discussing key elements, challenges and resources needed to facilitate cross-sectoral collaboration for age-friendly change.

## 2. Akita and Manchester: Age-Friendly Cities

Manchester, located in the north of England, is the largest city in the Greater Manchester region, itself the largest municipality outside of London. Once a global hub during the Industrial Revolution, since the decline of traditional manufacturing industries in the latter half of the twentieth century, Manchester has experienced a decline in investment and an increase in areas of high deprivation. Recent decades have benefitted from regeneration programmes and a growth in the tourism, sporting and educational industries. However, Manchester still has some of the areas of highest deprivation in England, and both life expectancy and healthy life expectancy are below national averages [23].

Manchester was the first city in the UK to join the WHO Global Network of AFCC. The ‘Age-Friendly Manchester’ programme is led by Manchester City Council (the local government) and is based in the Public Health department. Although Manchester is a relatively young city that is ageing slowly [14], many of the city’s older residents live in relative poverty and experience multi-morbidities. Therefore, in their work towards creating an age-friendly city, the programme works with many partners (internal and external to the City Council) in their aspirational efforts to make Manchester ‘a city for life’ [24]. Partners include third-sector organisations, many of whom work with specific communities of identity within the diverse Manchester population.

Akita is what has been termed a ‘super-aged city’, defined as a city with 21% or more of the population over 65 [25]. Population ageing in Akita is a result of increasing life expectancy, low fertility and outward migration from the city. The population of Akita over 60 is 39%, this represents a higher proportion than in Japan as a whole, which has the highest proportion of older people out of all countries globally. Situated in the northwest of Japan, Akita traditionally was an agricultural hub. However, as industry demands shifted in the twentieth century, the demographics of the city have changed. Now, many young people leave Akita to attend university or find employment in other cities and do not return. As a result, the population of Akita is decreasing, a trend that is expected to expand to the rest of the country, with the population of Japan predicted to decrease from 126 million to 100 million people between 2020 and 2050 [26].

Akita was an early member of the WHO Global Network of AFCC. The Akita Age-Friendly City programme is based in the Social Services department at Akita City Hall. The city is the centre of the regional Akita Prefecture. The rural nature of the prefecture creates many challenges for transport, particularly in the winter months when there is heavy snow. ‘Conceptualizing and creating an age-friendly city together, [working] towards a society where anyone can live a fulfilling and active life’ is a high priority for the city [27]. Challenges in the city relate to high levels of isolation and cases of dementia within the older population. Of all households in Akita, 20.5% are occupied by older people aged 65 or over living alone. The exact number of people living with dementia in Akita is unknown, but it is estimated to be about 25% of older people aged 65 and over [28]. To address these challenges, partnerships with community associations and private sector organisations are important to the Akita Age-Friendly City programme [29].

## 3. Methods

A case study approach was used to facilitate the generation of ‘an in-depth, multi-faceted understanding of a complex issue in its real-life context’ [30] (p. 1). This approach allowed the research to explore how cross-sectoral collaborations were mobilised to deliver AFCC programmes within the local context of each case study city. The case study design was particularly well-suited to this research, as it enabled both an analysis of commonalities and differences across cities and a deeper, context-specific understanding of localised practices. While cross-national studies of AFCC programmes remain limited, a multiple case study approach has proven effective in past studies for exploring how age-friendly policies and initiatives function in diverse settings [9,31].

### 3.1. Case Selection

Purposive sampling was used to select the cities based on three criteria: (1) a demonstrated, sustained commitment to ageing as a policy priority; (2) membership of the Global Network of AFCC and at least ten years of age-friendly policy development and implementation; (3) diversity in geographic location as well as cultural, socio-economic and demographic characteristics. The cities’ long-standing engagement in an AFCC programme was critical to ensure that they could both offer the depth of policy knowledge and the practical experiences necessary for a meaningful analysis of their successes and ongoing challenges. Access to key stakeholders was a crucial logistical consideration, as their active cooperation was vital for facilitating data collection and ensuring the research could be executed effectively [30]. A summary of the key characteristics of Akita and Manchester is presented in Table 1.

### 3.2. Data Collection

Between March 2022 and June 2024, the case studies were developed through a process involving (1) working with key stakeholders to create city profiles, (2) conducting semi-structured interviews across a wider group of stakeholders, and (3) carrying out fieldwork in each city, involving site visits and observations of age-friendly settings and initiatives. This research received full ethical approval from the University of Manchester’s ethics committee (Review Reference: 2021-11832-20386).

Firstly, to understand the context in each city, city profiles were created in collaboration with key local stakeholders, including AFCC programme leads, local government representatives, and academic experts researching ageing in the cities. Collecting data in collaboration with key local stakeholders allowed the research team to tap into expert contextual knowledge. Stakeholders were provided with a standardised city template to complete, which collected information on (1) urban characteristics (governance structure, the history of the city, economic change, relative wealth, and national context); (2) demographic data (ageing profile of the city, (healthy) life expectancy, comparisons with national trends, and diversity within the population); (3) Age-Friendly City development (the narrative of age-friendly work in each city, including the history, timeline, WHO Global Network status, key actors, mechanisms for working with older people, and examples of age-friendly policies and initiatives in each city). Stakeholders were also asked to provide links to data sources for further follow-up by the research team; additional data sources included local government reports, census data, social data, academic publications, and age-friendly policy documents. The research team developed the final city profiles based on the templates and drew on the additional data sources to ensure consistency and comprehension across both cases.

Secondly, building on the city profiles, semi-structured interviews were conducted with key stakeholders (Akita *n* = 8, Manchester *n* = 9). Purposive sampling was used to select participants, ensuring the representation of a diverse range of actors involved in the age-friendly programmes in each city. While some of the interviewees had already contributed to the city profiles, the interviews offered an opportunity to further explore specific themes, clarify initial data, and gather deeper insights into the barriers and enablers to delivering age-friendly policies and initiatives. The sample included local government employees, older volunteers supporting age-friendly programmes, representatives from community organisations, academics, and private sector stakeholders (Table 2).

The interviews were guided by a semi-structured interview schedule (see Appendix A), which was pilot-tested in February of 2022 and included questions related to the research questions while allowing participants to discuss issues they felt were important. Key topics covered were the priorities of age-friendly work, the governance and delivery structures, examples of age-friendly policies and initiatives, and the challenges and barriers faced. Most interviews in Manchester were conducted in person (except for two, which were conducted via Zoom). Interviews with participants in Akita were conducted via Zoom with the assistance of an interpreter. The Japanese research team checked the quality of the translations and ensured that the meaning and cultural context of the data were understood. Interviews lasted between 50 min and 2 h, with those using an interpreter lasting the longest. All interviews were audio recorded with the permission of the participant and transcribed by a third party. Interview data were managed using NVivo 12.

Thirdly, UK team members conducted site visits and participant observation of age-friendly initiatives to deepen their understanding of how the work is implemented on the ground. In Akita, one week of fieldwork involved visits to intergenerational projects, community-led social activities, health workshops, local government offices, and collaborative initiatives involving public and private sector partners. Each visit, lasting between two and four hours, was facilitated by local stakeholders and selected for its relevance to age-friendly programme implementation and collaboration. During these visits, UK team members engaged in conversations with participants and stakeholders, asked questions about programme collaboration, and observed interactions and programme delivery. These observations were enriched by the prior experience and insights of the Japanese team members, who had long-standing involvement in Akita’s age-friendly initiatives.

In Manchester, the UK research team drew on their extensive prior involvement in key age-friendly initiatives, including a range of neighbourhood-based research projects [32,33,34]. While no new fieldwork was conducted in Manchester for this study, insights from previous fieldwork experiences and existing field notes provided deep contextual knowledge of the city’s age-friendly programme development. These prior experiences, and those of the Akita-based research team, combined with site visits and observations in Akita, enriched the case study development and contributed to a more grounded and nuanced understanding of each city’s AF ecosystem. Although the fieldwork observations were not the primary data source, they provided valuable context that enriched the analysis and helped to contextualise data gathered through other methods.

### 3.3. Data Analysis

Data analysis followed an iterative process. Firstly, the completed city profiles were read in detail and discussed to identify similarities, differences and key aspects of cross-sectoral collaboration. This included mapping the range of actors involved in the respective age-friendly programmes (spanning local government, the private sector, community organisations, academia and older people) and identifying examples of age-friendly initiatives delivered through cross-sectoral collaboration.

Secondly, using the city profiles as a foundation, a priori codes were developed as a starting point for deductive thematic coding of the interview transcripts. Coding was carried out by UK-based researchers who independently read transcripts before comparing codes for consistency [35]. Initial codes included age-friendly actors, approaches to age-friendly work, collaborative approaches, challenges, city context, and co-production with older people (Table 3). This initial phase was followed by refinement of the coding through inductive analysis, focusing specifically on the mechanisms that drive cross-sectoral collaboration among actors. During this stage, initial codes were explored for recurring patterns and thematic connections within the data, resulting in three overarching themes. For example, codes related to ‘age-friendly actors’ and ‘collaborative approaches’ and ‘challenges’ coalesced into a broader theme of leadership and influence as a key driver of collaboration. At this stage, contextual observations from the fieldwork visits, combined with the research team’s prior knowledge, informed the case study analysis and were triangulated with the findings from the city profiles and interviews to corroborate the final themes. The emerging themes were subject to review by the wider research team and refined through a process of consensus. The final themes identified were leadership and influence, co-production and place-based working. How the themes were defined in the context of the data is discussed within the findings.

## 4. Findings

The first research question sought to understand the range of stakeholders working in the age-friendly cities of Akita and Manchester. Both cities have age-friendly programmes led by the local government, and they work with stakeholders in health services, community organisations and the private sector, as well as older people themselves, but these dynamics vary across the cities. Akita works very closely with the private sector, while Manchester has a stronger relationship with community-based organisations. Manchester has support from a regional and national network of AFCC programmes, while Akita has limited support from outside of the city for its age-friendly work. The size of the older population varies significantly across the cities, and, while Akita, as a super-aged city, has ageing issues high on the city’s agenda, Manchester works closely with its Older People Board in order to continually raise awareness of the growing and diverse needs of older people. This study primarily focused on how these diverse actors interact and collaborate, with the roles of the various actors highlighted throughout the findings.

The second research question explored what mechanisms drive cross-sectoral collaboration among diverse actors in age-friendly cities. Analysis of the Akita and Manchester case studies resulted in the following three themes relating to the mobilisation of cross-sectoral collaboration in age-friendly programmes: (1) leadership and influencing, (2) co-production, and (3) place-based working. Each of these will be explored in more detail. The themes are not mutually exclusive, and the insights from the case studies demonstrate how multiple mechanisms are often used to mobilise collaboration. A summary of the themes and examples of how different actors are engaged in cross-sectoral collaboration across Manchester and Akita is provided in Table 4.

### 4.1. Leadership and Influencing

When exploring the mechanisms for mobilising cross-sectoral collaborations in creating age-friendly cities, ‘leadership and influencing’ was a recurring theme in both case studies. Beer and Clower [36] argue that defining leadership is reliant on context, but that place-based leadership is often collaborative as opposed to hierarchal; it can be informal or formal, and it aims to achieve sustainability of communities over time. In our analysis, we defined a leader as someone who aligns diverse interests fosters a shared vision and sustains momentum for age-friendly change [37]. The following examples illustrate how leaders in the context of AFCC programmes were able to influence others to mobilise cross-sectoral collaborations.

Within Manchester City Council, the age-friendly agenda is led by a small team that has fluctuated in size over the years in line with government funding changes and budget cuts. While the team based in the Public Health department leads the age-friendly programme, delivering services across the AFCC framework’s domains relies on influencing and building partnerships to gain support from other local government officers. One way influence has been achieved is through strategic partnership building and identifying where priorities overlap. An example is the partnership between the Manchester (local government) and Greater Manchester (regional government) age-friendly programmes. As put by a local government stakeholder, when both programmes were renewing their strategic plans, ‘*we said*, “*Right, we’ll work in parallel, in partnership, and develop our strategies in parallel with that*”. *Where there’s opportunity to work together*, *where there’s crossover, we’ll obviously look at how we can do that*’ *(Local government, Manchester)*. Collaboration across local and regional governments helps raise awareness of their mutual policy priorities on ageing, a particular challenge in Manchester given its high levels of deprivation and need and relatively small older population (Table 1).

In Akita City Hall, the age-friendly programme is also led by a small team. However, while the age-friendly programme is an established unit within the city, the leadership of the programme has routinely changed over time. The age profile of Akita (Table 1) means that (unlike in Manchester) leaders do not need to spend as much time and effort in keeping ageing on the agenda. However, within the ageing agenda, there are different approaches and priorities (such as a stronger focus on dementia care in Akita), and the age-friendly team needs to work to influence across the City Hall and external stakeholders to increase awareness of and support for the age-friendly agenda. This effort includes distinct branding and the promotion of the ‘Akita Age-Friendly City’ logo, which is used as a pin badge worn by City Hall staff and displayed by businesses and services designated as ‘age-friendly’ across the city. These businesses earn the designation by meeting specific criteria, such as training staff on age-friendly practices and providing amenities like seating for customers in need of rest.

The success of this cross-departmental influence is evident in the integration of the age-friendly agenda within city-wide plans: ‘*We have action plans with other departments involved as well and we have some measures to take. So then they will report to us, other departments, according to these action plans that we set up*’ *(Local government, Akita)*. For example, the 2016–2020 Akita Age-Friendly Action Plan included an indicator to ‘continue to improve the outdoor spaces, buildings and facilities’, with actions led by the Health and Welfare Department, the Gardens Division and the Akita Prefecture Police.

In contrast to Akita, despite some changes in the Age-Friendly Manchester team, the case study revealed strong and consistent leadership that has sustained age-friendly work over many years. Evidence of the success of the strong leadership can be seen in the range of partnerships external to the city council and the extent to which Age-Friendly Manchester has been ‘*championed by a whole range of people external to the programme*’ *(Community organisation, Manchester)*. For example, Age-Friendly Manchester initiated a project in a neighbourhood south of the city centre that mapped the social and physical infrastructure of the area and aimed to identify ways of improving environmental features to support ageing in place [38]. The project received funding from the local housing provider, who also provided a dedicated older people’s project officer. As described by an academic stakeholder (also a partner on the project), since then they have had ‘*a ten-year relationship with [the housing provider] on various initiatives and it’s been really nice to see them develop their programme over that period of time and have that real institutional commitment to doing [age-friendly work]*’ *(Academic, Manchester)*. At the end of the project, the findings were used to consider the implications for other districts across the city.

In both cities, influence and leadership have led to the age-friendly work being championed by the respective mayors, a significant achievement that supports the mobilisation of age-friendly work across internal and external partners. However, age-friendly leaders need to continually foster partnerships and influence to ensure that election promises and policy pledges are followed through. Here, older people play a key role in leading the age-friendly agenda and holding local government officials to account. Evidence from both cities shows that charismatic older people, acting as influencers, collaborate with age-friendly programmes as leaders and advocates for the age-friendly agenda, particularly through Age-Friendly boards. One stakeholder in Manchester noted the following:

We’ve got some … really strong activists. So, for example, the chair of the Age Friendly board, she will be [engaging with communities] across the city … she’s been involved in the training with us. She’ll be at the launch tomorrow … she’s always there.(Community organisation, Manchester)

Examples of similar dedication and activism by older people were evident in Akita. However, Akita was often described as a conservative city, influenced by its agriculture-based industrial past and geographic distance from major urban centres. This environment fostered a mindset where the saying ‘*the nail that sticks out gets hammered down*’ *(Private sector, Akita)* reflects the challenges of advocating for change, as it sometimes requires one to “stick out”. In contrast, Manchester is a city that prides itself on its innovation and industrial culture. The slogan ‘Doing Ageing Differently’, used by the regional government, highlights an acceptance, or even an expectation, of diverse styles of leadership that promote creativity in developing age-friendly initiatives.

In summary, leaders, whether from local government, older people or community organisations, play a key role in becoming age-friendly influencers and forming and maintaining age-friendly collaborations. A significant barrier to age-friendly leadership is the turnover of key personnel over time. As mentioned, the size of the Manchester team has fluctuated over time, while the Akita team routinely experiences personnel changes. To counter these fluctuations, a key aspect of leadership is ensuring sustainable collaborations and partnerships that are resilient enough to withstand changes within age-friendly programme teams.

### 4.2. Co-Production

The second theme identified as a mechanism of cross-sectoral collaboration was co-production, defined as the active involvement of older people in shaping initiatives that directly impact their lives and communities [39]. Co-production serves as a vital mechanism of cross-sectoral collaboration by facilitating partnerships among diverse stakeholders—such as local governments, private sector entities, community organisations *and* older people—while fostering shared responsibility in the development of age-friendly initiatives. In particular, this approach empowers older individuals, transforming them from ‘passive beneficiaries’ into ‘active participants and leaders’.

Age-Friendly Living Lab Akita (ALL-A) is an example which demonstrates how co-production can facilitate cross-sectoral collaboration. Established in 2019, ALL-A receives support through investments and seconded staff from three private sector businesses: Cable Networks Akita, Akita Bank, and Akita Sakigake Shimpo (Local Newspaper). By contributing staff and resources to ALL-A, these businesses view their involvement as a social contribution to the city. The aim of ALL-A is to improve the happiness and well-being of Akita’s super-aged population by connecting people and companies. To achieve this, ALL-A engages with older people to collectively nurture community-building efforts. The lab works with a core group of 306 (January 2024 figure) older people who are signed up as members and who help lead workshops and run classes. For example, they hold ‘culture classes’ where they teach skills such as using computers or smartphones. ALL-A collaborates directly with the Akita Age-Friendly City programme to tailor their support. For example, they have put in place workshops to follow up on the specific needs of older people identified through public health checks. These workshops involve learning about how to manage frailty and create a self-management plan to prevent the onset of symptoms of frailty. As described by one stakeholder, ALL-A is ‘*taking measures together with industry, academia, and government to provide continuous preventative measures for older people*’ *(Private company, Akita)*. Thus, the workshops delivered by ALL-A are co-produced, and the multi-sectoral collaborations are sustained through their shared goal of exploring ways to adapt to and support the super-aged population in Akita.

A further illustration of co-production as a mechanism for cross-sectoral collaboration is the ‘Ambition for Ageing’ programme from Manchester. The Ambition for Ageing programme was part of a large seven-year initiative aimed at supporting people to age in place across Greater Manchester. Multiple stakeholders were involved; the programme was delivered by a not-for-profit community-based organisation, informed by research through a partnership with The University of Manchester and was strategically influenced by the Age-Friendly Manchester programme. One stakeholder put it as follows:

Ambition for Ageing was part of … the Age-friendly Strategy, the city council had elements in it around neighbourhoods and about creating more age-friendly neighbourhoods, and so the actual delivery of the work was part of [Age-Friendly Manchester] achieving their strategic outcomes. (Community organisation, Manchester)

Therefore, the different partners worked collectively towards the shared goal of supporting people to age in place, and each partner benefited in a way that supported their organisational objectives: the community organisation received funding to deliver community-based services, the university received evidence to support their research, and the city council (through the age-friendly programme) could demonstrate actions meeting their strategic plans.

In the UK, there is an increasing demand for political transparency and demonstratable examples of responding to the needs of residents. A co-production approach helps Age-Friendly Manchester to meet this expectation by working with older people in their programme delivery, specifically through the Older People’s Board, which is now an established mechanism for transferring decision-making power to older people. The mutual benefit of this approach is illustrated in the following quote:

I think that [older people’s] views are being taken more seriously now, and they’re sitting on more boards, and there’s more older people more confident to speak out. And because of the kind of public service reform agenda, and the fact that it’s like, “We need to hear: this is what you said you wanted, this is what we did”, I think that’s coming … I think it can be really hard to get it right, but I feel like there’s much more of a culture of doing that now.(Community organisation, Manchester)

The above quote reflects that, although the process is working, it is not an easy process; it is well established that time and resources need to be put into co-production to achieve genuine outcomes [32]. As raised by another stakeholder, “*there’s still this dependence on the people who’ve got that opportunity to engage and hear, to have that power to engage or not*” *(Local government, Manchester)*. Therefore, there is a recognition within the Age-Friendly Manchester programme that there is not yet equality of opportunity for older people to engage in working with the programme.

The challenges of using co-production were also evident from the Akita case study. Akita does not have an older people’s board in the same way as Manchester but rather has an age-friendly committee made up of a range of stakeholders external to the city hall, including older people. Therefore, the committee is a cross-sectoral collaboration, and its role is to support the age-friendly programme, but there were different expectations of how the committee should work. As put by one member, ‘*I always had the feeling that maybe the activities that they were carrying out are not as effective as I would like them to be. And because it’s a public—it’s a city hall, so they are not as flexible as private companies*’ *(Older person, Akita)*. The above comment reflects how co-production, and cross-sectoral collaboration more generally, are challenged by managing diverse expectations and ways of working, particularly in relation to effectiveness and flexibility.

### 4.3. Place-Based Working

To achieve age-friendly change in neighbourhoods, it is vital to work with community organisations, community leaders and local businesses that understand the needs of older people and the wider community in the areas they serve [40]. Findings from both the Akita and Manchester case studies suggest that place-based working is an effective mechanism for cross-sectoral collaboration. Many barriers associated with urban living are experienced by older people in their immediate neighbourhood, including lack of public transport, inaccessible footpaths, limited public seating and insufficient meeting places [39]. By leveraging the local knowledge and expertise of local organisations, leaders and businesses, age-friendly programmes can adopt a place-based approach, directly addressing the specific needs of older people in their neighbourhoods.

One existing mechanism of place-based working is the ‘Neighbourhood Coordination Groups’ operating across Manchester. These groups facilitate the tailoring of services to the ethnic, cultural and socio-economic diversity of older people across Manchester, specifically at the neighbourhood level. Although all neighbourhoods represent a mix of people from different backgrounds and experiences, there is clustering across the city of various groups (including students, ethnic minorities and lower-income households) in specific areas. By collaborating with the Neighbourhood Coordination Groups, Age-Friendly Manchester can access a range of place-based local government officers, housing providers and voluntary sector organisations to meet the needs of older residents in different areas. As put by a local government stakeholder, through these coordination groups, ‘*we link neighbourhood-based workers into the work of the [Age-Friendly] programme and strategic priorities, but [also] we learn about what they’re doing to inform our policy*’ *(Local government, Manchester)*. In summary, working through a place-based approach allows for a response that reflects the heterogeneity of the older population at the local level.

The strength of the place-based approach was demonstrated during the lockdowns associated with the COVID-19 pandemic. Across Manchester, existing community-based organisations were able to connect the needs of their communities with support from public services and communicate public health messages in ways that resonated with their communities [40]. One stakeholder talked about how supporting ageing in place ‘*means cooperation between statutory, voluntary and health services*’ (*Community organisation, Manchester)*. However, they went on to explain that because of the restructuring of public services and funding, place-based working has historically faced challenges. Nonetheless, the pandemic highlighted the value of partnerships and place-based working: ‘*I think through Covid, local working relationships—where people … know about our project—[enabled partnership working at] … the neighbourhood level*” *(Community organisation, Manchester)*. This quote demonstrates how, when a need was felt most keenly, priority was given to local partnerships and place-based services that were able to meet the needs of the older residents in their communities.

Reflecting a similar place-based approach, the Akita Age-Friendly City programme works in partnership with the neighbourhood-based chounaikai (neighbourhood associations) to deliver many of the age-friendly services. Chounaikai are long-established community organisations that serve as mechanisms for local coordination and support. Often run by older volunteers, these associations facilitate communication among residents, promote social engagement and address the specific needs of communities. However, the effectiveness of chounaikai varies significantly. As one stakeholder noted, “*some are functioning still, and some are not …, but if it is functioning, obviously they can get more support. For example, if they [older volunteers] live in the same place for a long time, they know each other, so they can go and check on somebody. So that [this support] will happen if their neighbourhood association is functioning*”. The crucial role that older people play in sustaining the chounaikai and supporting their peers in the community was highlighted as particularly vital. This engagement not only fosters a sense of community but also contributes to a broader sense of *ikigai*, a concept in Japanese culture which embodies purpose and fulfilment in life, as it provides opportunities for older residents to connect, contribute and find meaning through volunteering; this was explained by one older person in the following way:

The priority for me is ikigai. It’s a difficult word to translate. For elderly people to have something that they feel worth living for. For example, a place they get together. So, like maybe one or two places in each town for elderly people to get together to do something, so they feel like they’re part of a society.(Older person, Akita.)

While many chounaikai exist, not all have a physical place to meet, which presents a particular challenge in the more rural areas of the city. To address this, the local government provides several Comprehensive Support Centres across the city, offering a variety of services and activities. This was explained by one stakeholder as follows:

There are eighteen centres in Akita City at the moment, and it depends on the area, the priority is slightly different. In some areas we have to prioritise how to tackle financial issues or transport issues, [in other areas] people are more worried about dementia. So, it depends on the area, the priority is slightly different.(Community organisation, Akita.)

Therefore, the Comprehensive Support Centres provide vital social infrastructure, defined physical and organisational structures that support the social needs of communities, for delivering age-friendly services. Within these centres, older people play a leading volunteer role in service delivery. An ongoing challenge, therefore, is to ensure that there are older people willing to take on these leadership roles within their communities. It is also crucial to sustain these roles over time, ensuring that when current leaders can no longer serve, others are prepared to step in and continue the work.

## 5. Discussion

Although it is well established that the development and delivery of AFCC require cross-sectoral collaborations across a range of actors, less is known about *how* these collaborations are mobilised. Across the two case studies explored in this paper, a diverse range of actors engaged in fostering age-friendly cities. Through an in-depth comparative case study analysis of Akita in Japan and Manchester in the UK, three mechanisms that mobilise cross-sectoral collaboration to support the implementation of age-friendly programmes have been identified: Leadership and influencing, co-production, and place-based working. These mechanisms are essential for mobilising a diverse range of actors within the age-friendly eco-system.

Both cities delivered their age-friendly programmes through local government teams, engaging support from various departments and working with older people to inform their respective initiatives. This partnership building and maintaining of relationships were facilitated by effective leadership and influencing. With both programmes running for many years, evidence suggests a cumulative effect of age-friendly leadership and influence, resulting in a sustained embedding of age-friendly messages within the wider agenda of the cities. However, when leadership changes, including shifts in external political leadership, there is a risk that this influence may lessen. In Manchester, the relationship between local and regional government is crucial for ‘keeping ageing on the agenda’, further supported by the national UK Network of AFCC. In contrast, Akita lacks a similar regional or national programme and relies solely on its own local infrastructure to maintain momentum for its age-friendly initiative. Meanwhile, the growing number of national age-friendly programmes, as part of the Global Network of AFCC, highlights the importance of support and resources from national networks in ensuring that ageing remains a priority on local agendas [41]. However, for the age-friendly agenda to have a meaningful impact, leaders within the Global Network of AFCC, as well as regional and national affiliates, must navigate the challenge of integrating age-friendly issues into broader societal agendas, such as tackling climate change, addressing inequalities, and promoting sustainable urban development, ensuring that ageing is considered as a central element in policy and planning across these areas [8].

The significant role that older people have played in leading the age-friendly agenda has created a situation where there is ‘no going back’ to earlier times when their engagement and participation were not valued. Co-production is increasingly recognised as an effective tool for empowering older residents to assume leadership roles within age-friendly programmes, particularly through their participation in older people’s boards, as seen in both Akita and Manchester. However, to ensure equitable access to these leadership opportunities, it is essential for age-friendly programmes to reflect on who is involved—and especially who is not—and to consider how potential barriers to participation may be addressed. This includes diversifying leadership to represent a broader range of voices, backgrounds and experiences, as well as implementing systems that compensate older individuals for their time and contributions, recognising their invaluable input in the co-production of age-friendly initiatives [33]. Further, the Akita case study highlighted the essential role that older people play as volunteers; however, the sustainability of relying on an increasingly older volunteer workforce needs to be examined to ensure that services can be delivered to all who need them.

Another way to examine cross-sectoral working is to question who is *not* involved in age-friendly collaborations. Valuable insights can be gained from Akita’s approach to working with the private sector to support its super-aged population. Recognising the role of older people as one of their prime consumers, local businesses have partnered with the age-friendly programme to deliver tailored age-friendly services, and, in the case of Age-Friendly Living Lab Akita (ALL-A), large local businesses are seconding staff to better understand and support the needs of the growing older population. This level of engagement with the private sector is not as evident in Manchester. While several Manchester stakeholders discussed attempts at partnership building with private housing companies, these relationships resulted in mixed levels of commitment towards creating age-friendly change. Perhaps these relationships would be stronger if greater emphasis was placed on the mutual benefits that partners would gain from engaging with age-friendly programmes (as in Akita). In both cities, there was evidence of working with public health sectors, but more could be done to bring health and social care services into the age-friendly ecosystem [11].

Across both case studies, the vital role of community-based networks (for example, the neighbourhood coordination groups in Manchester) and community associations (for example, the chounaikai in Akita) in mobilising collaboration through place-based working was clear. To ensure the sustainability of age-friendly initiatives at the local level, it is essential to allocate more resources to these community organisations. As local governments face increasing financial pressures while trying to support their ageing populations on reduced budgets, it is likely that there will be greater reliance on community-based organisations. Yarker highlights the uneven capacity of neighbourhoods to support the diverse needs of ageing populations, suggesting that, without adequate and equitable funding, place-based organisations will be unable to fulfil their role in supporting the development of age-friendly initiatives [42]. The lack of sufficient resources not only limits the effectiveness of these organisations but also undermines the long-term viability of age-friendly initiatives, potentially relegating them to temporary, piecemeal efforts rather than sustainable, systemic change [15].

Finally, the multiple roles played by older people in both case studies demonstrate their key contributions and leadership within age-friendly programmes [43]. Older people as participants are essential to the fabric of the age-friendly city, contributing their insights and experiences to shape inclusive environments. At the same time, older people as volunteers provide critical support that underpins the success of these programmes. However, it is older people as influencers who challenge age-friendly programmes to deliver services that meet the diverse range of needs of both ageing and aged communities. The *leadership roles* taken on by older people range from serving on age-friendly boards to acting as local activists who deliver place-based services, thereby fostering a sense of purpose (ikigai). While many older people participate as volunteers, influencers, and leaders, there are still challenges in fully integrating their voices into decision-making processes on age-friendly initiatives. Future research is needed to explore strategies for enhancing and sustaining participation and leadership from older people in the age-friendly agenda, particularly focusing on involving diverse groups to ensure that all voices are represented.

### Limitations

There were several limitations to the study outlined in this paper. First, both case studies focus on cities that have been delivering age-friendly programmes for some time. As a result, the mechanisms for collaboration may differ in cities that are new to the age-friendly approach. Second, while we compared two distinct geo–socio–political environments, both are situated within relatively wealthy economies. Cities in the Global South are likely to utilise different mechanisms of collaboration due to variations in macro-level governance and micro-level family support structures [44]. Third, the development of these case studies was informed in part by the perspectives of actors within the respective age-friendly programmes, which may introduce some bias in how their successes are portrayed. Additionally, the need to conduct interviews online during the COVID-19 pandemic limited our ability to engage with a wider range of perspectives. Further research could benefit from exploring more perspectives from actors outside of age-friendly programmes, examining a broader range of contexts, as well as understanding which strategies drive change in resource-limited cities, and how age-friendly goals can be aligned with critical issues such as poverty alleviation, healthcare access, and urban migration.

## 6. Conclusions

Leadership and influencing, co-production, and place-based working are interconnected mechanisms that play a crucial role in fostering cross-sectoral collaboration to create age-friendly communities. While co-production and place-based working are vital for delivering age-friendly initiatives, they fundamentally rely on strong leadership and influence to thrive. It is, therefore, essential that all three mechanisms are leveraged to effectively mobilise cross-sectoral collaborations in age-friendly programmes. We recommend that a range of actors are proactively engaged in the delivery of AFCC programmes—local governments, private businesses, community organisations—to collectively create age-friendly environments to support healthy ageing. Furthermore, as cities transition from ‘ageing’ to ‘aged’ populations, greater emphasis should be placed on recognising and integrating the diverse experiences of ageing individuals, along with the various formal and informal roles they play in shaping age-friendly initiatives and supporting healthy ageing. Supporting these diverse efforts will not only broaden the relevance of age-friendly initiatives but also ensure that the voices and aspirations of older individuals are at the forefront of shaping age-friendly futures.

## Figures and Tables

**Table 1 ijerph-22-00073-t001:** Key characteristics of case study cities.

	AKITA	MANCHESTER
**POPULATION**	307,672	552,858
**POPULATION AGED 60 AND OVER**	39%	13%
**LIFE EXPECTANCY (MALE/FEMALE)**	80.5/86.9	75.5/79.9
**IMMIGRANTS AS A PERCENTAGE OF TOTAL POPULATION**	0.4%	31.4%
**OECD AGEING CATEGORY** ^1^	Super-aged society	Ageing society
**JOINED GLOBAL NETWORK OF AGE-FRIENDLY CITIES AND COMMUNITIES**	2012	2010

^1^ Classifies cities by the proportion of older people aged 65 years or more [25].

**Table 2 ijerph-22-00073-t002:** Summary of interview participants.

		AKITA	MANCHESTER
**NUMBER OF INTERVIEWS**		6	9
**TRANSLATOR USED** yes/no		6/0	0/9
**IN-PERSON/ON-LINE**		0/6	7/2
**NUMBER OF PARTICIPANTS**		8	9
**ROLE:**	Local government	2	2
	Older people	1	1
	Community organisations	3	4
	Private sector	1	0
	Academics	1	2

**Table 3 ijerph-22-00073-t003:** Summary of top-level thematic analysis of interviews.

Code	Description	Example
**Age-friendly actors**	Participants discussing other actors: local government, the private sector, community organisations, academia and older people	“We have these initiatives to create partnerships with companies, not like government companies but private companies, who agree with policies to create an age-friendly city” (Local government participant, Akita)
**Approaches to age-friendly working**	Systems and mechanisms for age-friendly policy development and delivery	“Having that evidence-based approach is something that you really value … I think that just gives a bit more weight to things and I like to work like that, I like to have that evidence” (Local government participant, Manchester)
**Collaborative approaches**	Partnerships and cross-sectoral working,—how collaborations happen	“Members of committees are usually like, for example, university professors or public servants who have been involved in care work or welfare, and people from ageing, we call them ageing people’s clubs … and we also ask the general public to come and join … and also somebody like myself from the media are also members of this committee”. (Private sector, Akita)
**Challenges**	Challenges associated with cross-sectoral collaboration	“One of the big things I come back to is shifting of minds. We got to work more on that. … It all comes down to influence and policies and procedures and help to initiate change”. (Older person, Manchester)
**City Context**	Demographics, urban context, attitudes towards ageing and older people	“The young people leave home and move to big cities, so the elderly people have to support each other, that is the sort of system that we have been working with already, but we will need this more in the future.” (Community organisation, Akita)
**Co-production with older people**	Including the voice of older people, working with the people who the policy affects	“The assembly, which is the open group to anybody over fifty in the city, I think that’s really important. That’s that wider voice of older people, which is then reflected in the fact that a proportion of the [older people’s] board is elected from that forum”. (Local government, Manchester)

**Table 4 ijerph-22-00073-t004:** Summary of themes and examples of actors engaged in cross-sectoral collaboration across Manchester and Akita.

Theme	Examples of How Different Actors (in Bold) Are Engaged in Cross-Sectoral Collaboration	
	Akita	Manchester
**Leadership and influencing**	The AFCC programme is based in the **Social Services Department** and uses **cross-departmental** influence to ensure the integration of the age-friendly agenda within city-wide plans.	The AFCC programme is led by the **Public Health Department** in local government but relies on strategic partnership building and identifying where priorities overlap with **other local government departments**.
	To increase awareness of the age-friendly agenda, **local businesses** can commit to criteria and be designated as ‘age-friendly’ businesses.	A strong **programme leader** has been credited for maintaining the range of partnerships external to the city council.
	**Older people**—acting as influencers—sit on the multi-stakeholder **Age-Friendly Committee** as leaders and advocates for the age-friendly agenda.	**Older people** play a key role in leading the age-friendly agenda and holding local government officials to account through the **Age-Friendly Older People’s Board**.
**Co-production**	**Age-Friendly Living Lab Akita (ALL-A)** is a community-based organisation that links older people with companies, receiving support through investments and seconded staff from three **private sector businesses**.	**Community-based organisations** work with the **AFCC programme** to deliver projects, such as ‘Ambition for Ageing’, which aimed to support people to age in place.
	**ALL-A** collaborates directly with the **AFCC programme** to tailor their support.	Community-based age-friendly work is frequently informed by **researchers**, and strategically influenced by the **AFCC programme**.
	An **Age-Friendly Committee** made up of a range of stakeholders external to the city hall, including **researchers**, the **private sector** and **older people**.	The **Older People’s Board** is an established mechanism for transferring decision making power to **older people.**
**Place-based working**	The AFCC programme works in partnership with the neighbourhood-based **chounaikai (neighbourhood associations)**—often run by **older volunteers**—to deliver many of the age-friendly services.	By collaborating with **Neighbourhood Coordination Groups**, the AFCC programme connects a range of place-based **local government officers**, **housing providers**, and **voluntary sector organisations** to address the needs of older residents in different areas.
	The **local government** provides several **Comprehensive Support Centres** across the city, offering a variety of services and activities.	During the COVID-19 pandemic, **community-based organisations** linked community needs with **public service support** and effectively communicated public health messages.
	**Older people** play a vital role as **volunteers** in delivering place-based services and activities.	A place-based approach allows for a response that reflects the significant **heterogeneity of the older population** at the local level.

## Data Availability

See Appendix A for data sources.

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
