# Peer review of "Mobilising Cross-Sectoral Collaboration in Creating Age-Friendly Cities: Case Studies from Akita and Manchester"

_ijerph, 2025, doi:10.3390/ijerph22010073_

Round 1

Reviewer 1 Report

Comments and Suggestions for Authors

Thank you for the invitation to review this paper, which analyses the mechanisms that drive collaboration in two Age Friendly Cities and Communities (Manchester and Akita). I am not an expert in the field of AFCC, but can provide the view of a social scientist interested in ageing studies more generally. The paper is excellently written, clear, and engaging. I found very little that needed improvement. I’ve included a few suggestions here, but these are not essential.

Lines 37-42 – the grammar here is a little confusing – it reads as if the concept was developed through a study exploring the key characteristics of AFCC? I suggest rewording.

L57-64 – this conceptualisation might work better in the earlier paragraph

L62 – Buffel and Phillipson’s (missing ‘s)

L140 – a couple of words to immediately define ‘super-aged city’ would be useful

L195 – the primary research institution (Manchester?)

L212 – suggest changing ‘drawing’ to ‘drew’

L225 – It would be useful for readers to be able to see the interview schedule

L235 - change comma to full stop

L284-297 – findings relating to the first research question (one paragraph) are treated lightly compared to the findings of the second research question (4.5 pages). I wondered if more could be said about the range of stakeholders, for example through diagrams showing the networks in the two cities. This is not essential but would allow a visual comparison, if these data are available.

L380 – I wondered if the concepts of champions, opinion leaders and agents of change might be relevant here – for example in Roger’s Diffusions of Innovations.

L568 – who’s quote is this?

L575 – it is not clear here what is meant by ‘no going back’ – this needs a little more discussion or rewording.

L639 – ‘actors outside of age-friendly programmes’ – and in a wider range of contexts, as you say earlier in the paragraph? 

Thanks again, and best of wishes with the paper.

Reviewer 2 Report

Comments and Suggestions for Authors

Dear authors,

The topic you addressed is very important and provocative in the context of ageing and urbanisation, and your work in this area is very important for advancing our understanding.

Please find some comments/suggestions you could consider to improve the paper's clarity.

The manuscript's title is impactful. The abstract is well-written and adheres to the IMRAD format for scientific papers. However, the meaning of "cross-sectoral collaboration" in the context of the case studies remains unclear.

The paper's introduction establishes a foundational context for the research. The authors could extend/ enrich this section by elaborating more in-depth on the criteria and rationale behind selecting two distinct cities with varying demographic characteristics. 

It would be beneficial to clarify the concept of "age-friendly cities," as this term is referenced but not thoroughly defined. 

Additionally, emphasising the significance of highlighting these particular cities within the research will enhance the reader's understanding of their relevance to the study's goals.

 A critical perspective on 'age-friendly cities' is needed. The assertion that the two towns are committed to age-friendly initiatives is not sufficiently supported. 

What criteria did you use to select the key stakeholders?

Are there differences in the terminology used in the various programs? What commonalities exist among these cities, and how do their 'age-friendly cities' programs differ? 

Do the three themes arise solely from the analysis of the interviews, or do they also originate from the 'age-friendly cities' programs?

Analysing age-friendly city programs could yield more interesting findings; additional sections would help clarify the paper's ideas.

A diagram illustrating the relationships among the three themes would provide a more comprehensive understanding of the issue addressed.

The discussion section is well-developed;  however, the paper must include more insights into the three themes. It favours a more descriptive approach rather than an epistemological one.

This emphasis on description limits the potential for a more profound epistemological analysis, which could enhance the reader's comprehension and engagement with the material. Shifting towards integrating these insights could significantly improve the overall argument and depth of the paper.

Reviewer 3 Report

Comments and Suggestions for Authors

General Comments

Thank you for your contribution. The article draws on examples from a comparative case study across Akita (Japan) and Manchester (UK), two cities with distinct demographic profiles but both with a longstanding commitment to the age-friendly approach. Especially about cross-sectoral collaboration, and to date there is limited understanding of what mechanisms and strategies drive collaboration among diverse actors within age-friendly cities.

Introduction

Point 1: The introduction provides a thorough background but repeats information about AFCC's importance several times.

Point 2: The research gap is identified but could be more explicitly stated.

Methods

Point 3: The methodology is detailed and aligns well with the study objectives. However, some procedural details, such as the interview question design are missing. Could author provide an appendix with sample interview questions to enhance transparency?

Point 4: The justification for case selection is valid but could include additional comparative elements like cultural or political contexts.

Point 5: How were the observations in Akita validated or triangulated with Manchester, given the differences in data collection methods?

Findings

Point 6: The findings are well-structured under clear themes, but some sections are overly descriptive. Could author use tables or bullet points to summarize key similarities and differences across the two cities?

Point 7: There is limited direct comparison between Akita and Manchester in some parts. Is there any discussion on how cultural factors influenced the success or challenges of the mechanisms in each city?

Discussion

Point 8: The discussion effectively ties the findings to existing literature but lacks critical reflection on potential biases. How can the identified mechanisms be adapted for cities in developing economies with fewer resources? Could author discuss the potential challenges of scaling these mechanisms to national or regional levels?

Conclusion

Point 9: It does not reflect on the study's limitations or propose future research directions explicitly. Would including specific recommendations for stakeholder groups (e.g., local governments, private sector, community organizations) strengthen the conclusion?
